# Myopericarditis Associated with COVID-19 in a Pediatric Patient with Kidney Failure Receiving Hemodialysis

**DOI:** 10.3390/pathogens10040486

**Published:** 2021-04-17

**Authors:** Marcela Daniela Ionescu, Mihaela Balgradean, Catalin Gabriel Cirstoveanu, Ioana Balgradean, Loredana Ionela Popa, Carmen Pavelescu, Andrei Capitanescu, Elena Camelia Berghea, Cristina Filip

**Affiliations:** 1Department of Pediatrics, “Carol Davila” University of Medicine and Pharmacy, 020021 Bucharest, Romania; daniela.ionescu@umfcd.ro (M.D.I.); mihaela.balgradean@umfcd.ro (M.B.); catalin.cirstoveanu@umfcd.ro (C.G.C.); loredana.popa@drd.umfcd.ro (L.I.P.); camelia.berghea@umfcd.ro (E.C.B.); 2“Maria Sklodowska Curie” Emergency Children’s Hospital, 041451 Bucharest, Romania; carmen.pavelescu@rez.umfcd.ro (C.P.); andreicapitanescu@gmail.com (A.C.); 3Department of Pharmacy, George Emil Palade-University of Medicine, Science and Technology of Targu Mures, 540142 Tirgu Mures, Romania; ioanabalgradean1@gmail.com

**Keywords:** end-stage kidney disease, SARS-CoV-2, myopericarditis, pleurisy, children, mortality, prevalence, hemodialysis, post-infective complications

## Abstract

The outbreak of COVID-19 can be associated with cardiac and pulmonary involvement and is emerging as one of the most significant and life-threatening complications in patients with kidney failure receiving hemodialysis. Here, we report a critically ill case of a 13-year-old female patient with acute pericarditis and bilateral pleurisy, screened positive for SARS-CoV-2 RT-PCR, presented with high fever, frequent dry cough, and dyspnea with tachypnea. COVID-19-induced myopericarditis has been noted to be a complication in patients with concomitant kidney failure with replacement therapy (KFRT). This article brings information in the light of our case experience, suggesting that the direct effect of severe acute respiratory syndrome coronavirus 2 (SARS-CoV-2) infection on cardiac tissue was a significant contributor to myopericarditis in our patient. Further studies in this direction are required, as such associations have thus far been reported.

## 1. Introduction

Children and adolescents with COVID-19 fare considerably better than adults, with mortality rates in pediatric patients (age < 18 years) of less than 1% reported in early studies [1,2,3]. Although pediatric cases of COVID-19 are less severe than in adults, a small number of children with underlying conditions such as chronic kidney disease (CKD) and hypertension may increase the risk of severe illness from COVID-19 at any age, knowing that those cells that express angiotensin-converting enzyme 2 (ACE2) (including lung cells, kidney, and cardiovascular tissues) are at risk for SARS-CoV-2 infection [4,5,6].

SARS-CoV-2, the causative agent of coronavirus disease 2019 (COVID-19), is known to cause a spectrum of symptomatic infection, which can progress to pneumonia and acute respiratory distress syndrome. Cardiac involvement associated with COVID-19 has been recently described in the literature, and may include myocardial injury, cardiomyopathy, myopericarditis and cardiac tamponade [7,8,9]. Myocarditis is known as the inflammation of the muscular walls of the heart, and remains a diagnostic challenge in the clinical context due to the variability of presentations in the pediatric population. It has significant morbidity potential, including decreased cardiac function and cardiac failure [10].

Pericarditis is an inflammatory condition that affects the sac surrounding the heart, which is most often due to viral infections. Poor prognostic factors include the presence of a large pericardial effusion, tamponade, myopericarditis, and high CRP or NT-pro-BNP [11].

SARS-CoV-2 utilizes the angiotensin-converting enzyme 2 (ACE2) receptor, which is abundant in the lower respiratory tract, for entry into the cells [12]. ACE2 is highly expressed in many extrapulmonary tissues, including heart, kidney, endothelium, and intestine, making all of these organs potential targets [13,14]. The expression of the ACE2 receptor can cause renal involvement, and severe renal dysfunction is more common among patients with chronic comorbid conditions, especially patients with CKD. After the initial infection, the acute disease progression can be divided into three distinct phases—an early infection phase, a pulmonary phase, and a hyperinflammation phase which is characterized by a cytokine storm, leading to immune-mediated injuries to distant organs [15].

## 2. Case Presentation

A 13-year-old girl, known with kidney failure with replacement therapy (KFRT), was referred to the Nephrology and Dialysis Department of the Emergency Hospital for Children “Maria Sklodowska Curie” of Bucharest, Romania, for high fever (38.5–39 °C), chills, frequent dry cough, and dyspnea with tachypnea on 9 July 2020.

The patient has a medical history of CKD since 2015, secondary to polycystic kidney disease (PKD), which is the most common genetic disorder leading to dialysis dependence. Autosomal dominant polycystic kidney disease (ADPKD), the most common cause of KFRT, was diagnosed in our case on abdominal CT, showing multiple bilateral renal cysts. In this case, we report a pediatric patient with a family history of PKD. The study of the prevalence and causes of kidney imaging correlates with the abdominal imaging from her father, who also has PKD. The images of the brain revealed Bourneville tuberous sclerosis (TSC) with cortical dysplasia (diagnosed at the age of 6 months). The concurrence of TSC and ADPKD is rare, with very few cases reported in the literature. Hypertension is a typical manifestation of ADKPD and the cause of the rapid progression of CKD to KFRT. The complex relationship between TSC, epilepsy, and autism spectrum disorder (ASD) leads to high risk in this category of individuals, and was confirmed in the Neurological Department. In December 2019 she began receiving chronic intermittent hemodialysis, 3 sessions weekly, 4 h per session. She was under chronic treatment with enalapril 20 mg/day and amlodipine 10 mg/day, with partial control of blood pressure values. In June 2020, echocardiography revealed severe left ventricle hypertrophy, normal systolic function of both ventricles, left ventricle diastolic dysfunction (impaired relaxation), and mild pulmonary hypertension.

At admission on 9 July 2020, the patient was vitally stable, with fever of 39 °C, a respiratory rate of 35 to 40 breaths/min, oxygen saturation of 80% to 85% in ambient air, and SpO2 was 95% with oxygen 4 L/min, heart rate of 120 beats/min, blood pressure (BP) of 160/110 mmHg, and oliguria (500 mL/24 h). The physical examination revealed a sick patient with dyspnea, tachypnea, decreased vesicular murmur on the right hemithorax, fine bilateral crackles and wheezing, diminished heart sound, systolic murmur, abdominal distension due to enlarged kidney, and peripheral edema.

Based on the COVID-19 outbreak, a nasopharyngeal swab was performed, with a positive result on the following day on real-time reverse transcriptase-polymerase chain reaction assay for SARS-CoV-2. The patient was diagnosed with COVID-19. Her condition worsened over the following few days, and the general condition was modified. During hospitalization, investigators revealed hypoalbuminemia, anemia, and mild inflammatory syndrome, and NT pro-BNP was 32,460 pg/mL (see Table 1).

On 15 July 2020, imaging investigations detected severe pericardial effusion, bilateral pleural effusion, and ascites (20 mm pericardium, 40 mm abdominal, 30 mm left pleura, 20 mm right pleura). Echocardiography revealed moderate/severe systolic dysfunction of the left ventricle, impaired diastolic dysfunction (pseudonormal transmitral pattern), low-normal right ventricle systolic function, moderate pulmonary systolic pressure, and a large amount of pericardial fluid. The clinical picture maintained despite supplemental daily hemodialysis sessions performed with maximal ultrafiltration (13 mL/kg/h) [16,17]. She was transferred to a COVID-19 designated hospital for isolation and specific treatment. The patient responded gradually to medical therapy (antihypertensive drugs, high diuretic doses, daily hemodialysis sessions), and her oxygenation continued to improve until she was successfully weaned from oxygen. Her systolic and diastolic LV function gradually improved and she could be discharged in good clinical condition.

After 2 months from COVID-19 infection, on 22 September 2020, the patient returned to our clinic for frequent dry cough, dyspnea with tachypnea, and respiratory rate 25 breaths/minute, without fever. RT-PCR test was negative. The patient had bilateral basal crackling rales, more common on the left side, tachycardia AV-120 bpm, hypertension (BP—145/95 mmHg), Sat O2 = 80%, and diuresis 1000 mL/24 h. Radiological images during this hospitalization can be seen in Figure 1A–C.

The echocardiography indicated severe LV systolic dysfunction (EF = 25%), severe diastolic dysfunction (restrictive transmitral pattern), grade II mitral regurgitation, mild RV systolic dysfunction (EF = 45%), moderate/severe pulmonary hypertension, and a large amount of pericardial fluid. (See Figure 2, Figure 3, Figure 4, Figure 5 and Figure 6).

The patient was started on broad-spectrum antibiotics with intravenous ceftriaxone (50–100 mg/kg/day) 2 g/day, and vancomycin (500 mg/day, intravenously, single daily dose—dose adjusted for renal function, treatment maintained for 14 days), intravenous methylprednisolone pulse therapy (2 mg/kg/day, for 3 days). She also followed treatment with high doses of diuretic (furosemide 120 mg/day), inotropic therapy (dobutamine 5 µg/kg/min), angiotensin-converting enzyme (ACE) inhibitor (enalapril 20 mg/day), calcium channel blocker (amlodipine 10 mg/day), and intensive hemodialysis programme (see Figure 7).

Her clinical condition progressively improved. After 2 weeks from her September hospitalization, she presented with SpO2 = 95%, partially controlled blood pressure, small pericardial and right pleura effusion, regression of pulmonary hypertension, and improvement of systolic function of the left ventricle (LVEF = 45% at patient discharge).

All procedures performed were in accordance with the ethical standards of the institutional and/or national research committee(s) and with the Declaration of Helsinki (as revised in 2013). Written informed consent was obtained from the mother of our patient.

## 3. Discussions

Cardiac involvement with myopericarditis is a possible late phenomenon in COVID-19 progression, but it is also suspected to be the primary reason of death associated to CKD risk mortality. To the best of our knowledge, a few myopericarditis cases in COVID-19 in pediatric patients have been previously described [18,19,20].

Pediatric hemodialysis patients represent a high-risk group for COVID-19 for several reasons. First, they cannot strictly quarantine at home due to the need for hemodialysis treatment several times weekly [21,22]. As the pathophysiology of COVID-19 infection is not completely understood, especially in patients with severe multiple organ dysfunction associated with myopericarditis, more attention should be given to patients with highly increased values of NT-proBNP and KFRT. Specific laboratory tests can be used to predict a worsening clinical course. The NT-proBNP level was very high (70,000 pg/dL), significantly increased compared to the acute episode of COVID-19 infection (32,460 pg/dL), (see Figure 8).

The patient presented slightly elevated cardiac troponin level (cTnT = 66 pg/mL), and D-dimer was 1.15 µg/mL. (see Table 1). Myocarditis in our patient occurs as a pre-existing cardiovascular disease, in the context of kidney failure, left ventricular hypertrophy, and diastolic dysfunction. Troponin values can be elevated in patients with COVID-19 infection with CKD, and elevations of other cardiac biomarkers (NT-proBNP, D-dimer) will be more specific as indicators of cardiac injury [23,24,25]. Two months after COVID-19 infection was confirmed, our patient exhibited severe systolic and diastolic dysfunction of left ventricle (more severe than cardiac decompensation from the acute phase of COVID-19 infection) associated with severe pulmonary hypertension and mild right ventricle systolic dysfunction.

The etiology of cardiac dysfunction in this case may be multifactorial. Direct cardiac injury may occur due to viral invasion, while the cytokine storm induced by COVID-19 may also have toxic effects on the myopericardium. We strongly considered that severe cardiac decompensation could not be explained only by chronic renal failure and pneumonia, in the condition of no evidence of sepsis. The cardiac improvement was probably due to specific cardiac treatment (preload/afterload decreasing by diuretics/vasodilators, intensive dialysis, and increase of contractility by inotropic medication) and to pulse-therapy with methylprednisolone with inhibition of post-COVID-19 systemic inflammatory response.

In this case, it is essential to emphasize the importance of echocardiographic diagnosis (using complementary techniques such as tissue doppler imaging (TDI), which can clarify the contribution of possible heart damage (systolic and diastolic) to the clinical picture dominated by dyspnea and cough so that it is not attributed only to the intrinsic respiratory component.

At the beginning of the pandemic, some retrospective reports suggested poor prognosis in patients infected with SARS-CoV-2 and RAAS blockade usage [26], but this was not confirmed in a meta-analysis [27]. Moreover, there were proved protective effects of these drugs, making ACE-I discontinuation not recommended [28], so we decided to continue with the previously established chronic antihypertensive therapeutic regimen including these drugs.

Out of the need to standardize the terms used in the scientific community dealing with kidney disease, we followed the recommendations in *Pediatric Nephrology*, where the publication of the recent nomenclature of the Consensus Conference on Improving Global Outcomes of Kidney Disease (KDIGO) led to the standardization of scientific terms. Following these aspects of the nomenclature, we used the terminology corresponding to the degree of renal impairment in our patient [29].

Additionally, elevated levels of NT-proBNP should be interpreted with caution, as in patients with advanced chronic kidney disease the renal clearance of natriuretic peptides (and especially NT-proBNP) is reduced, leading to increased serum levels. However, the evolution curve of the serum values of this parameter interpreted in a clinical context may have diagnostic value and may be useful for assessing the response to treatment and the evolution from a cardiological point of view [30], (see Figure 7).

## 4. Conclusions

Findings in the literature highlight the high mortality of individuals with underlying kidney disease and severe COVID-19, underscoring the importance of identifying safe and effective COVID-19 therapies in this vulnerable population.

The main limitation is the inability to draw conclusions on the basis of one case of myopericarditis in a patient with COVID-19, where the causality was not clearly established, and COVID-19-induced myopericarditis was the diagnosis of exclusion. In this case, the patient improved during hospitalization, although the long-term effects of the myocardial injury after COVID-19 infection are still to be determined.

However, this is the first reported case of myopericardial involvement in COVID-19 with associated kidney failure in a pediatric patient in Romania.

We want to share our findings, given the urgent need for correct diagnosis and therapeutic strategies to better manage COVID-19 patients and diminish the SARS-CoV-2 complications in critically ill pediatric patients.

## Figures and Tables

**Figure 1 pathogens-10-00486-f001:**
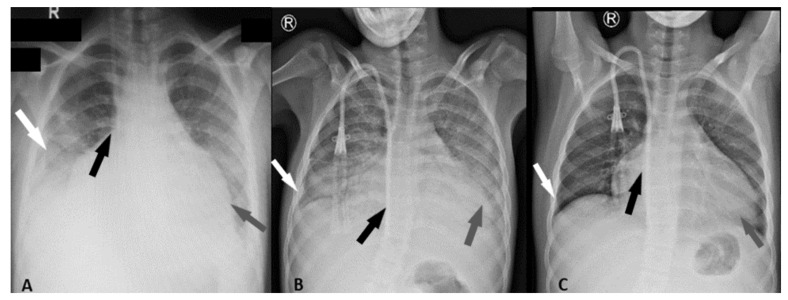
Serial radiological progression seen in a pediatric patient with COVID-19, CKD, and myopericarditis. (**A**) 15 July 2020—Chest X-ray (CXR) of COVID-19 patient on anterior–posterior projection shows patchy basal bilateral ground-glass opacities (GGOs), associating right reactive pleural effusion, and loss of lung markings in the mid and lower zones (white arrow). Increased transverse cardiac diameter possible by pericardial fluid effusion (grey arrow). Central venous catheter with right jugular insertion, paravertebral descending path, and internal extremity in the right atrium (black arrow). (**B**) 22 September 2020—Regression of alveolar infiltrates maintaining the consolidation from right inferior lobe; reduction of the pleural fluid reaction (white arrow), the globular enlargement of the heart shadow persists giving a water bottle configuration/cardio-mediastinal silhouette with increased transverse diameter (grey arrow), not significantly modified compared to the previous. Tunneled central venous catheter with right jugular insertion, paravertebral descending path and internal extremity at the level of inferior vena cava (black arrow). (**C**) 28 September 2020—Shows radiological improvement. An increase in bilateral normal basal pulmonary transparency; free lateral costodiaphragmatic sinuses (white arrow); the slightly increased diameter of the cardio-mediastinal silhouette is maintained, with the widening of the subcarinal angle (grey arrow).

**Figure 2 pathogens-10-00486-f002:**
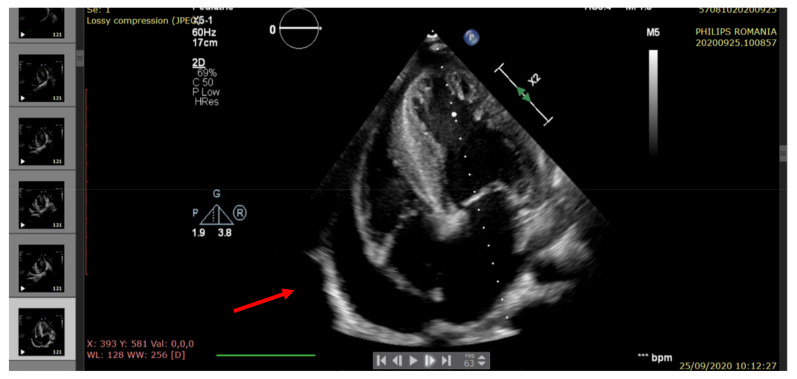
Echocardiography—apical 4-chamber view: a large amount of pericardial fluid (arrow), severe left ventricle hypertrophy (LVH).

**Figure 3 pathogens-10-00486-f003:**
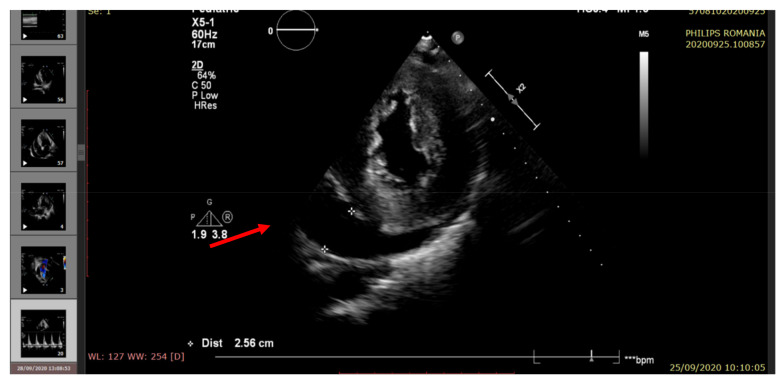
Echocardiography—parasternal short-axis view. A large amount of pericardial fluid (arrow), LVH.

**Figure 4 pathogens-10-00486-f004:**
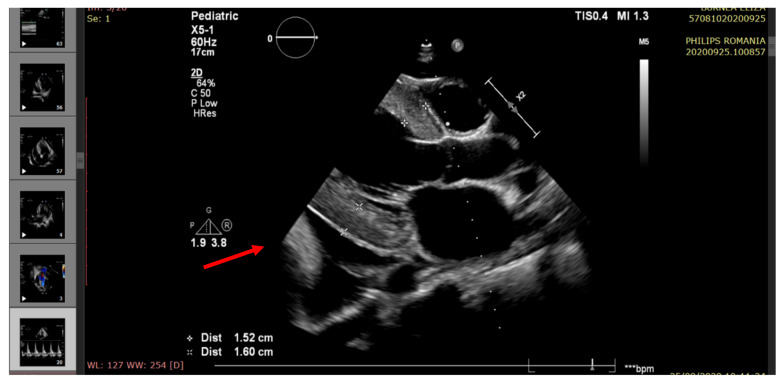
Echocardiography—parasternal long axis view. Posterior left ventricle pericardial fluid (arrow); severe LVH (LV mass index 185 g/m^2^).

**Figure 5 pathogens-10-00486-f005:**
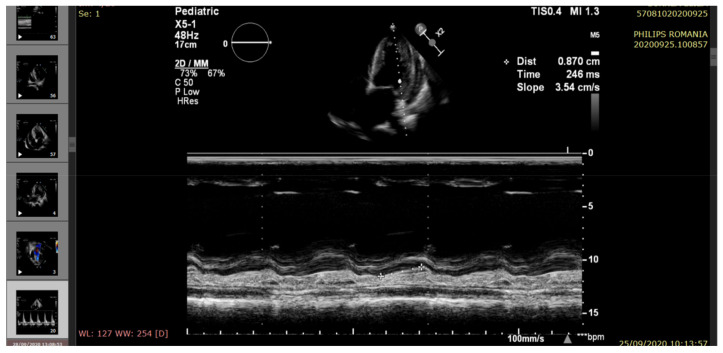
Echocardiography—apical 4-chamber view (M-mode). LV systolic dysfunction (MAPSE reduction).

**Figure 6 pathogens-10-00486-f006:**
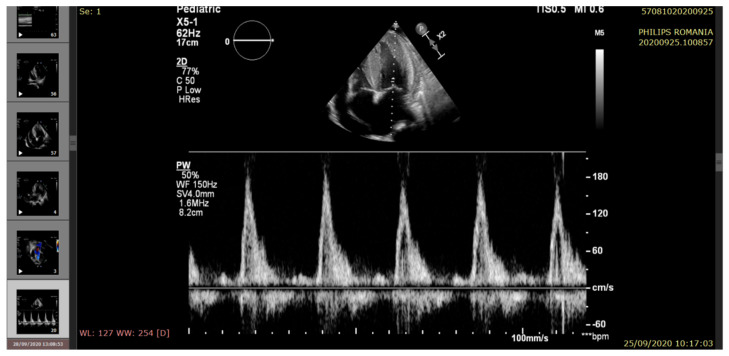
Echocardiography—apical 4-chamber view. HVS and restrictive diastolic pattern.

**Figure 7 pathogens-10-00486-f007:**
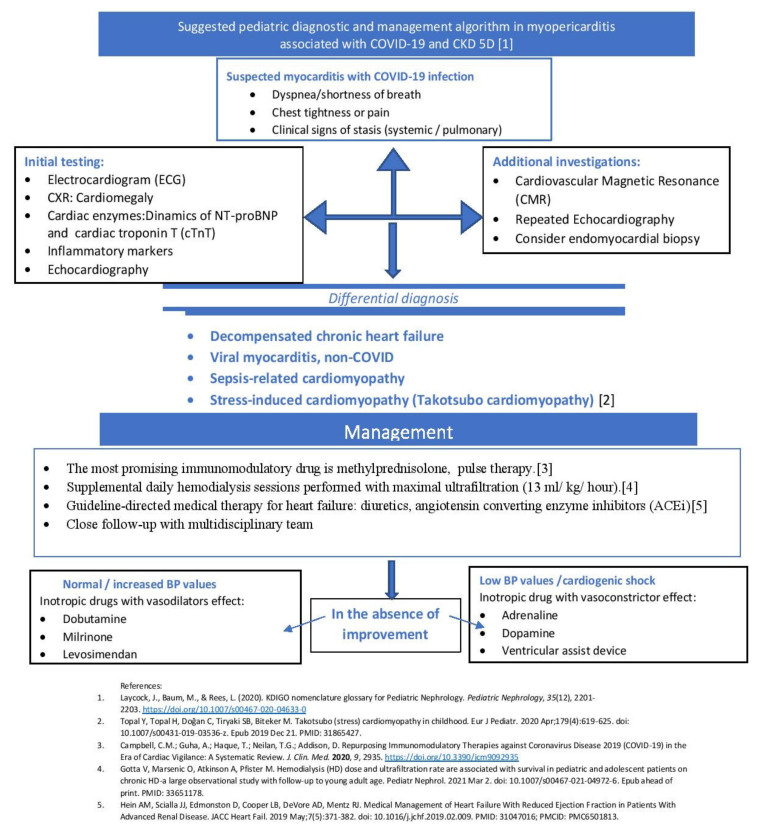
Suggested Pediatric diagnostic and management algorithm in myopericarditis associated with COVID-19 and CKD stage 5 treated by dialysis (CKD 5D).

**Figure 8 pathogens-10-00486-f008:**
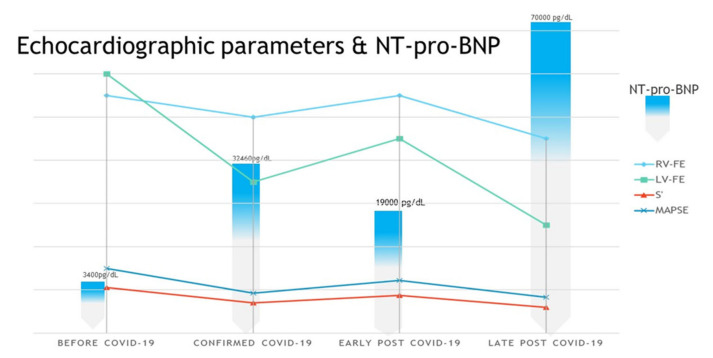
Longitudinal correlation of markers of cardiac injury (echocardiographic parameters and NT-proBNP values).

**Table 1 pathogens-10-00486-t001:** Echocardiographic data and essential laboratory findings at baseline (COVID-19 infection), before COVID-19, and early and late follow-up.

Parameter	Normal Range	Before COVID-19 (30 June 2020)	Confirmed COVID-19 Infection (9 July 2020)	Early Post-COVID-19 Infection (30 July 2020)	Late Post-COVID-19 Infection (22 July 2020)
Right ventricular ejection fraction RV EF (%)	>60	55	50	55	45
Left ventricular ejection fraction LV EF (%)	>60	60	35	45	25
(S’) Lateral systolic myocardial velocity (cm/s) tissue doppler imaging (TDI)	8.43 ± 1.06	10.5	7	8.7	5.9
Mitral annular plane systolic excursion (MAPSE) (mm) (M-mode)	16.4 ± 2.4	15	9.2	12.2	8.3
Mitral diastolic pattern	Normal	Impaired relaxation	Pseudonormal filling	Impaired relaxation	Restrictive filling
Pericardial fluid (edge)	<5 mm	Small (6 mm)	Raised (20 mm)	Moderate (14 mm)	Large (22 mm)
Pulmonary hypertension (echocardiographic criteria)	Normal	Mild	Moderate	Mild	Moderate/severe
Blood pressure (mmHg)	Systolic (90–120)/diastolic (50–80)	150/90	160/110	160/90	145/95
O_2_ %saturation	95–100	99	80–85	90	80
Heart rate (beats/min)	60–100	78	120	90	120
Diuresis (mL/24 h)	500–1200	1500	500	1500	1000
Na (mmol/L)	138–144	132	129.9	122	126.7
K (mmol/L)	3.4–4.9	4.90	4.96	5.00	5.02
Serum creatinine (mg/dL)	0.5–1	10.24	8.20	8.40	8.16
Serum urea (mg/dL)	<39	111.3	86.5	120	99.5
CRP (mg/L)	0–5.00	3.88	2.68	5.78	1.82
Hb (g/dL)	11.5–15	6.8	5.6	6.00	8.6
WBC (×10^9^/L)	4.5–13.5	4.43	5.67	6.57	16.33
Neutrophil (×10^9^/L)	1.8–8	2.5	2.65	16.00	14.14
Lymphocytes (×10^9^/L)	1.5–6.5	0.76	1.23	1.2	1.13
Lymphocytes (%)	20–55	17.20	6.1	6.00	6.9
NT-proBNP (pg/dL)	(<125)	3400	32,460	19,000	70,000
Troponin T (pg/mL)	0–14 (negative) >50 (positive)	23	61	44	66

## Data Availability

Not necessary.

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
