# Peer review of "Myopericarditis Associated with COVID-19 in a Pediatric Patient with Kidney Failure Receiving Hemodialysis"

_pathogens, 2021, doi:10.3390/pathogens10040486_

Round 1

Reviewer 1 Report

In this case report, Ionescu and colleagues reported a myopericarditis case that associated with COVID-19 in a pediatric patient with kidney failure receiving hemodialysis. This finding provides an evidence that SARS-CoV-2 infection might impairs cardiac tissue. Few things can be done to improve this manuscript quality for further publication.

Major comments:

1. Authors demonstrated that the patient was SARS-CoV-2 infection positive by Real-time PCR assay, on July 9, 2020. Would they provide the real-time PCR results, the CT value, as a main figure?

2. To help readers to understand the images that indicates the changes in lung and heart, authors might add few arrows or other labels in images to point the damage or changes in lung and heart. Also the description is needed in figure legends. 

3. In reviewer's opinion, show few critical parameters from Table 1 in a timeline figure could be more easier to read all the information in current Table 1. 

Minor things:

Make sure if a space is required between number and its unit.

In Table 1, a comma was used in some places for the parameter value.

Author Response

Dear reviewer, 

please find the answers in the document below. 

I hope we managed to meet your requirements correctly, we have attached the RT-PCR test.

Thank you,

Open Review 1

Major comments:

  1. Authors demonstrated that the patient was SARS-CoV-2 infection positive by Real-time PCR assay, on July 9, 2020. Would they provide the real-time PCR results, the CT value, as a main figure?

Answer: RT-PCR test in the attachments

  1. To help readers to understand the images that indicates the changes in lung and heart, authors might add few arrows or other labels in images to point the damage or changes in lung and heart. Also the description is needed in figure legends.

Answer: revision CP16 and 17, in the document, with added description. [CP18].

  1. In reviewer's opinion, show few critical parameters from Table 1 in a timeline figure could be more easier to read all the information in current Table 1.

Answer: a timeline with the critical parameters in COVID and myopericarditis and normal range in table 1, added. [CP12-16].

Minor things:

Make sure if a space is required between number and its unit.

Answer : Revision [CP12-CP16].

In Table 1, a comma was used in some places for the parameter value.

Answer: Revision [CP12].

Reviewer 2 Report

Please find my remarks in the enclosed pdf-file.

Note the punctuation and citation numbers positions.

Photos decriptions could be less sloppy.

Author Response

Please find the answers, below. I hope to meet your requirements.

Thank you,

Open rewiewer 2

1.Note the punctuation and citation numbers positions.

Answer: See revisions in the article: Removed 2 keywords, [CP3,4].[CP 9], [CP 16, CP 26, CP 28]

2.Photos decriptions could be less sloppy.

Answer: [CP19-20]

Reviewer 3 Report

This is a case report and some update would be justified  as the references need to  be updated since cardiomyopathy under covid 19 for children and adolescent  is now  much more reported 

Page 1 second paragraph : tamponnade shoudl be added to the list of cardiovascular compluication with a reference

Page 2 methylprednisolone need an "e" at the end 

Troponin values are not reported I guess this is because of kidney failure ,this should be explained.

A paragraph related to ACE enzyme shoul be written because this is the mostly pathway of cardiac dysfunction in this context  

Please enumerate normal values for echocardiac follow up (table)

Please annotate abnormalities in echocardiographic pictures 

Please resume and suggest specific protocol for patients having kidney failure and cardiomyopathy and COVID 19 infection which differentiate this presentation with others in the  same  field 

I  

Author Response

Please find my answers in the word document, in the attachment,

Thank you in advance,

Round 2

Reviewer 1 Report

Thanks authors' responses to my questions/comments. 

Reviewer 3 Report

The authors have improved the manuscript and responded to all my concerns